# Mass wasting reveals ongoing asymmetric retreat of the martian north polar ice cap

Shu Su [1] ✉, Lida Fanara [2], Haifeng Xiao [1,3], Ernst Hauber [2] & Jürgen Oberst[1]

Ongoing mass wasting through ice block falls is intensive at the north polar ice cap of Mars. We monitored how this activity is currently shaping the marginal steep scarps of the ice cap, which holds a record of the planet's climate history. With AI-driven change detection between multi-temporal high-resolution satellite images, we created a comprehensive map of mass wasting across the entire North Polar Layered Deposits (NPLD). Our results show a more active erosion process than previously thought, with scarps retreating by up to ~3 m every kiloyear. The distribution of the active scarps indicates an ongoing asymmetric retreat of the already subcircular ice cap. The active scarps and the interior dune fields correlate strongly with exposures of the underlying, sandier Basal Unit (BU), providing evidence that erosion of the BU undermines the base of the NPLD. Moreover, ice block fall activity suggests potential areas where gypsum is released, given that the interior gypsum-bearing dune fields are located adjacent to these active scarps. Here, our study reveals the rates of present-day topographic change of the north polar ice cap, providing a valuable constraint for study of its past evolution.

The martian north polar region hosts a prominent cap of water ice, built up from thousands of distinct layers[1,2]. It is like an old heavy book, with every page containing intriguing stories about the planet's past. Within the polar region, the so-called North Polar Layered Deposits (NPLD), composed of ~95% water ice and ~5% dust[3,4], are deposited over time on top of a sand-rich Basal Unit (BU). Both the NPLD and the BU have been reported to experience erosion, such as water ice ablation[5,6], frost-dust avalanches[7], and ice block falls[8,9]. These surface processes likely have been ongoing throughout the Late Amazonian epoch in response to orbitally-driven climate oscillations and polar geological processes[10–12].

It was long assumed that erosion of the bright layers of the BU and the lower NPLD is mainly due to sublimation[6,13–15], while later observations showed that massive mass wasting activity serves as the main contribution to erosion[8,12]. The equatorward-facing NPLD scarps can have slope angles up to 70° with obvious polygonal fractures (Fig. 1a). In particular, the ice block fall events, resulting from exfoliation fractures, occur frequently at the steep north polar scarps and are observed by repeat High Resolution Imaging Science Experiment

(HiRISE) coverage (Fig. 1b). Substantial amounts of water ice blocks rest at the feet of the steep scarps. In the past nearly 20 years it has become increasingly apparent that erosion due to ice block falls is frequent and widespread[8,9,16]. It is thought to be subject to internal/external triggers such as thermoelastic stresses[17], the seasonal $CO_2$ deposits[18,19], or removal of sand from the underlying layers by katabatic winds[20]. However, mass wasting has not yet been comprehensively quantified.

Following the release of high-resolution images of Mars, the ability to investigate meter-scale mass wasting activity has significantly increased the interest in assessing the retreat of icy scarps by analyzing ice block falls. Several studies[9,16] have provided estimates of mass wasting volumes, yet they were confined to individual scarps and limited time periods. Interestingly, studies on active NPLD scarps have also indicated exposed BU below[8,9,16,21] (subdivided into the rupes unit and the cavi unit[21], Fig. 1a). This raises the question of whether the exposed BU outcrops affect the mass wasting and steepening of the overlying NPLD scarps. Additionally, there is uncertainty regarding whether the observed mass wasting is sufficient to counteract the

[1]Institute of Geodesy and Geoinformation Science, Technical University of Berlin, Berlin, Germany. [2]Institute of Planetary Research, German Aerospace Center (DLR), Berlin, Germany. [3]Instituto de Astrofísica de Andalucía (IAA-CSIC), Granada, Spain. ✉e-mail: shu.su@campus.tu-berlin.de

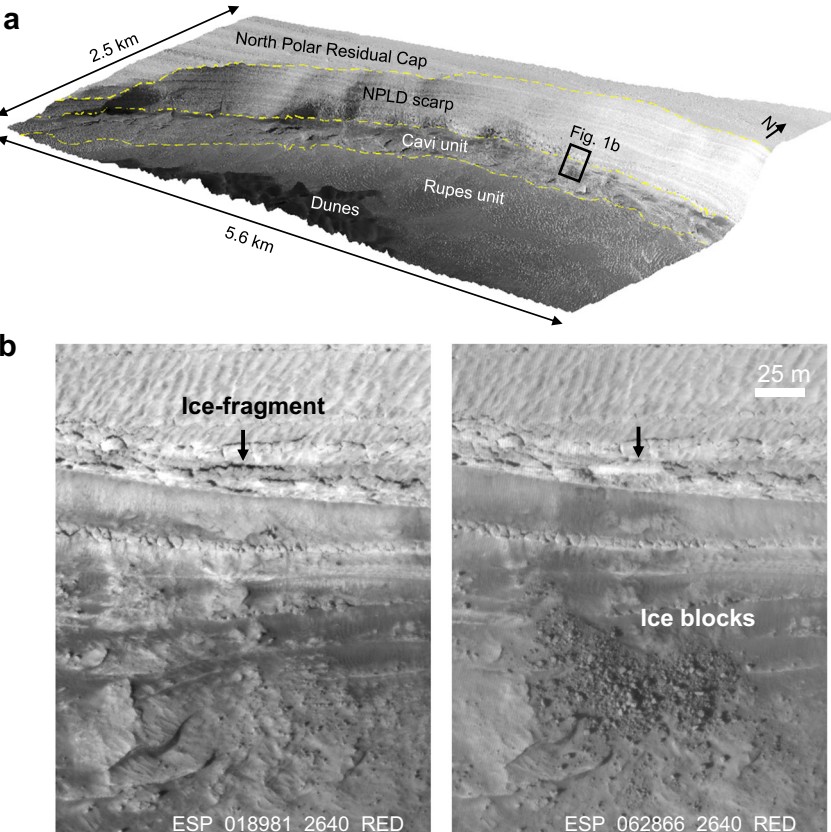

**Fig. 1 | Activity at the north polar region. a** A north polar scarp region, centered at 83.8°N, 236.2°E, in 3D view with 3× vertical exaggeration. The yellow dashed lines delineate the boundaries of each geological unit. **b** Multi-temporal HiRISE images show an ice-fragment detaching from the NPLD scarp surface (left) and breaking into a cluster of ice blocks at the foot of the scarp (right). Downhill is towards the bottom. The location is outlined in **a**. Credit: background HiRISE image in **a** and **b**, NASA/JPL/University of Arizona.

proposed viscous flow of the steep icy scarps that would have a shallowing effect to the scarp topography[22]. A previous study[9] estimated a minimum retreat rate of ~0.2 mm/yr for one scarp. However, a modeled viscous flow[22] showed a maximum flow rate of ~1 m/yr near the lower face of that scarp, which is 4 orders of magnitude higher than the estimated retreat rate[9]. Obtaining accurate retreat rate estimates for the entirety of the scarps is important for understanding the present state and evolution of the north polar scarps.

Early studies noted that some dune fields (Fig. 1a) lie close to the steep arcuate scarps, and they discussed the possibility that erosion of the BU supplied sand-sized materials to form the dune fields[20,23]. Further examination across the north polar region confirmed a strong positive spatial correlation of the dune fields and the BU outcrops[24]. Notably, gypsum, a hydrated calcium sulfate, has been predominantly discovered within dune covered regions, both in the Circumpolar Dune Field and the interior dune fields[15,25]. The polyhydrated sulfates found in the BU are suggested as the potential source of gypsum in Olympia Undae[26]. It is also suggested that the gypsum-bearing material is released from troughs and scarps by ice sublimation controlled by katabatic winds[6,15]. However, the source of these gypsum-bearing particles is still under discussion.

In this study, we utilize HiRISE images acquired to date to quantify the current NPLD mass wasting, specifically by detecting the sources of ice block falls, throughout the entire north polar region of Mars employing an advanced deep learning method[27]. The quantification of water ice loss from the active NPLD scarps facilitates the estimation of erosion and retreat rates. Monitoring such a substantial collection of scarps builds an invaluable pool of knowledge about the martian north polar ice cap and constitutes a key constraint in improving our understanding of its evolution.

## Results

### Comprehensive map of current mass wasting activity

We investigated all 2700 km of the north polar marginal scarps[15] and categorized them based on mass wasting activity and data availability (Supplementary Fig. 1). Then, deep learning change detection was applied on the scarps classified as active to delineate all detached ice-fragments (Methods). The estimation of the respective volumes over time resulted in the current erosion (Fig. 2a) and retreat rate (Fig. 2b) for each scarp.

The active NPLD scarps vary in length from ~8 km to ~80 km, with a minimal slope of 30° at their steepest point. These scarps are generally arcuate in plan-view, curving uniformly towards the north pole. Figure 2a shows that the vast majority of these active scarps are located within the longitudinal range of 110°E to 240°E. No active scarps have been found between 300°E–110°E, ~80°N. Notably, the degree of erosion varies between the scarps. Of particular interest are S2 and S9, which exhibit erosion rates of 0.88 and 0.66 $m^3$ per martian year per meter along the scarp, respectively. They are likely undergoing the most significant retreat of ~3 m/kyr (Fig. 2b). Only considering ice block falls, and assuming a density of $1200 \pm 200$ kg/$m^3$ for the NPLD[28], we estimate the mass wasting of the NPLD as ~$(9.2 \pm 1.5) \times 10^7$ kg/yr, with this number increasing if ice ablation is considered. The present-day accumulation rate of the NPLD has been inferred to vary from ~0.1 to 0.9 mm/yr, with associated net mass gain of ~0.2 to $1.8 \times 10^{12}$ kg/yr[29–32], i.e., greatly exceeding our estimated mass loss. However, estimated accumulation rates based on current data and models remain highly uncertain due to lack of constraints. Nevertheless, our study's quantification, based on observations of mass loss, could inform the models, thus, its use could extend beyond the results themselves.

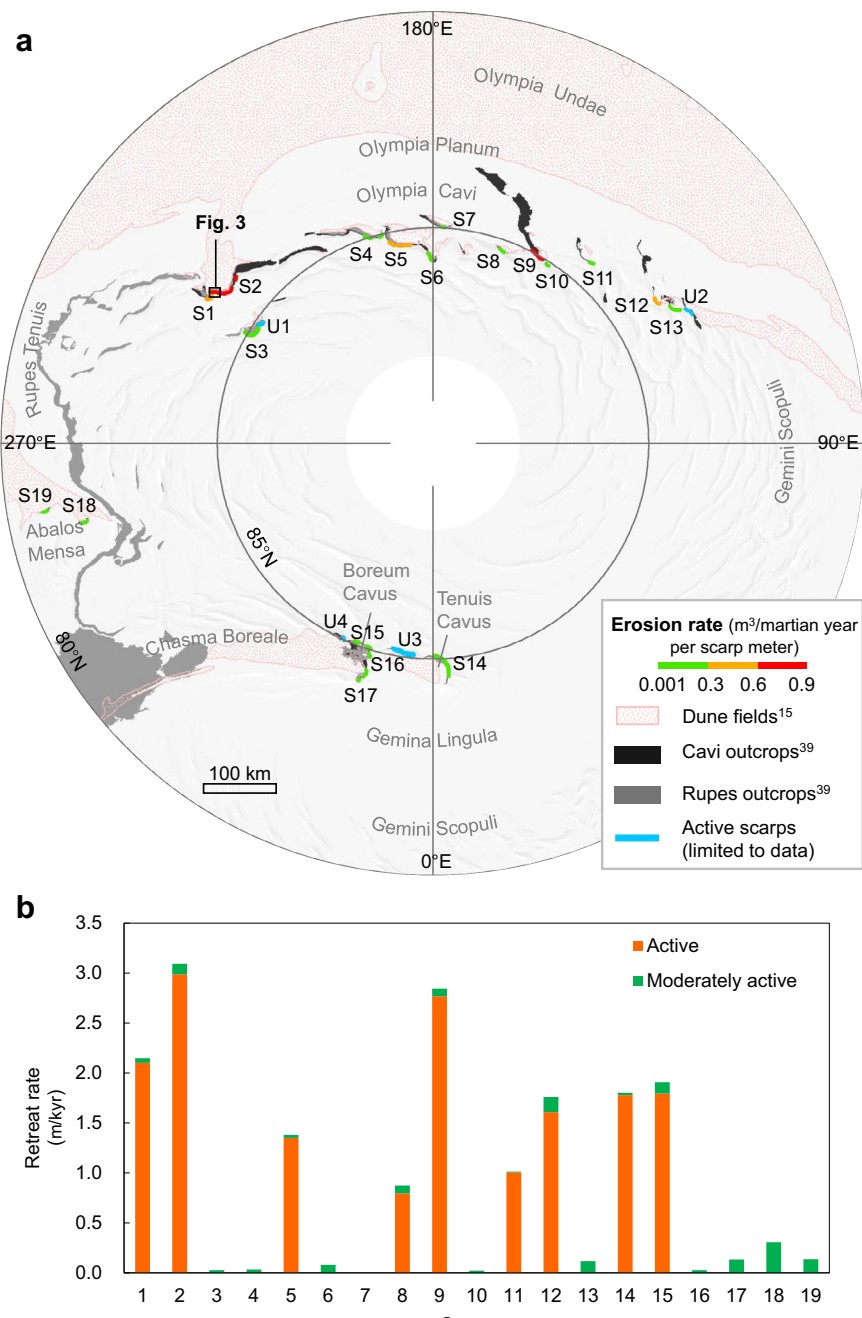

**Fig. 2 | Currently active NPLD scarps. a** Erosion rate of 19 scarps (S1–S19) across the north polar region. Scarps (U1–U4) are observed to be active with ice block falls, but the limited number of HiRISE images do not allow for change detection on these scarps. The lengths of the studied scarp sections are limited by the size of available HiRISE images. Dune fields are mapped by pink dots[15]. The cavi and rupes outcrops are mapped with black and gray patterns, respectively[39]. For a detailed map of cavi outcrops, rupes outcrops and dune fields, which are located close to S1–S19, please refer to Supplementary Figs. 8–26. The background is a Mars MGS MOLA Global Shaded Relief[46] map, in polar stereographic projection. **b** Retreat rates of the active and moderately active parts of the scarps.

## Scarp analysis highlights

Upon closer examination (Supplementary Figs. 8–26), we find non-uniform levels of mass wasting activity along the scarps both vertically and laterally. At S1 for example, approximately 85% of water ice has been shed from the lower part of the scarp, where the slope is much steeper (Supplementary Fig. 2). Moreover, S1 exhibits uneven retreat rates along the scarp. It is not uncommon to find parts of a single scarp heavily fractured while other parts remain inactive, usually at either end of the scarp (dark blue lines in Supplementary Fig. 1). Based on the activity level, we segmented each scarp into active parts, which typically exhibit severe fracturing, and moderately active parts where the scarp surface remains relatively intact with modest fracturing. Overall, we conclude that the more active lower parts define the retreat state of the scarps in comparison with the less active upper parts, e.g., with retreat rates of up to ~3 m/kyr (Fig. 2b).

The scarp S2 spanning ~37 km in length stands out as the most active scarp, with a measured ice loss of ~32,500 m³ per martian year (Supplementary Table 2) and a retreat rate of ~3 mm/yr at its lower part (Fig. 2b). To monitor mass wasting over time, a time series of change detection from MY 29 to MY 35 was performed on the part of S2 with the best image coverage. The results are visualized in the form of

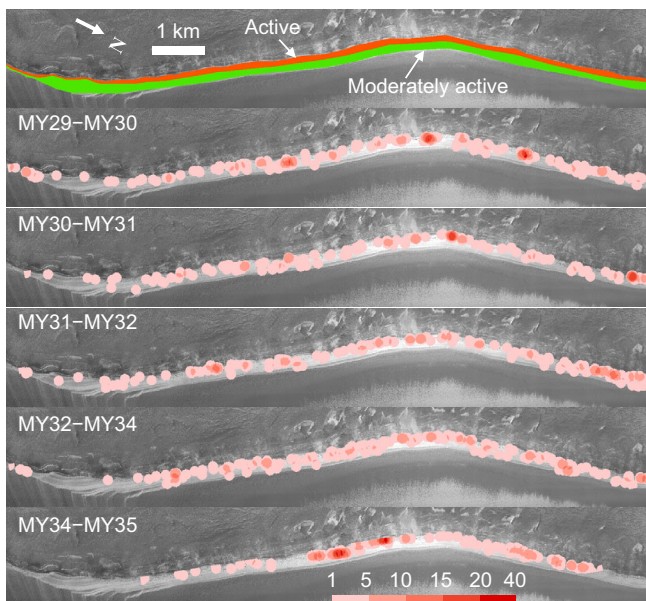

**Fig. 3 | Time series of mass wasting detection on part of the S2, outlined in Fig. 2a.** In the first image, the scarp is mapped into the active part (red) and the moderately active part (green) based on the fracturing level. The next five images are density maps of the detached ice-fragments at part of the S2, showing inter-annual results. Density is calculated as the number of detections in a circle with a radius of 100 m. Note that there is no HiRISE data during the summer of Mars Year (MY) 33. Downhill is towards the top. The background image is the red-filter HiRISE image ESP_019047_2640_RED. Credit: background HiRISE image, NASA/JPL/University of Arizona.

density maps with intervals of one or two martian years and reveal that the location of the intense activity changes over time (Fig. 3). However, the amount of ice loss is relatively similar each martian year, with no observed secular decrease or increase in ice block fall activity, indicating that mass wasting has been undergoing consistently over the past decade.

## Discussion
### Higher erosion rate than previously thought
Here, we present a comparison between our estimated rates of erosion and retreat with those in previous studies at the same S2 region (Supplementary Table 1). The detections by Fanara et al.[9] comprise ice block falls from both the NPLD and the BU scarps, which theoretically should yield higher values than our detections which focused solely on the NPLD scarps. Nevertheless, our retreat rate surpasses that of Fanara et al.[9] by an order of magnitude. The discrepancy most likely reflects the fact that ice-fragments breaking into ice blocks also produce an additional, substantial quantity of fine and pulverized materials that are not visible at HiRISE scale, and thus undetectable when searching for fallen material alone. This fact underscores the importance of detecting losses at the source region and the potential of comparing with the volume of ice blocks themselves to constrain the ice fragmentation process[16].

### Erosion vs viscous flow
For the same S2 region shown in Fig. 3, we estimate a retreat rate of ~3.6 mm/yr at its active part, which however cannot compete with the proposed viscous flow rate[22] of ~1 m/yr (Supplementary Table 1). By applying a flow rate of 1 m/yr over an approximately 12-year period (equivalent to ~6 martian years), the S2 base should have moved outward by ~12 m. However, when examining the orthorectified HiRISE images from this timeframe, there is no visible change (Supplementary

Fig. 3). We thus believe that the existence of such substantial viscous flow at the steep scarps is unlikely[33], and does not explain the existence of such a steep scarp with a slope up to 70 degrees. Uncertainties about the viscous flow modeling could arise because the dust fraction and its distribution in the layered deposits as well as the ice temperature are not well known[22,34]. From the analysis of S2 through time, we suggest a retreating state of all active scarps as they are undergoing frequent and significant mass wasting events, without any evidence of comparable viscous flow.

### Where have all the ice blocks gone?
Mass wasting activity probably has been undergoing for at least a couple of decades, or even possibly since the end deposition of the NPLD when the obliquity has been keeping below 30° and the ice deposits started reworking in response to climate change[10]. It is intriguing that the feet of the scarps are not covered by piles of ice blocks. One possible explanation could be that the ice blocks sublimate rapidly[35,36]. However, the rapid ablation of the fallen ice blocks is not evident in the multi-temporal images. The ice blocks may also be invisible in images due to fragmentation by thermal fatigue or being buried by large landslides. To test the above mentioned hypotheses, our work can serve as a constraint for both laboratory and modeling studies.

### NPLD activity is triggered by BU undercutting
All 23 active NPLD scarps are located directly above BU outcrops, particularly cavi outcrops, which are formed by interbedded bright icy layers and sand-sized lithic materials in dark tone[20] (Fig. 2a and Supplementary Table 2). The cavi unit is built typically above the rupes unit, which is composed mainly of fine-grained dust and water ice[28]. The NPLD scarps that lie directly above the rupes unit do not exhibit ice block falls (pink arrow in Supplementary Fig. 1). The discrepancy in composition between the cavi and the rupes unit shows that the cavi unit is more easily mobilized by aeolian processes, and thus more susceptible to erosion[20]. This increases the likelihood of the cavi unit undermining the NPLD base.

Each active NPLD scarp (except S10) is also associated with an interior dune field lying downslope within a distance of 0–10 km (Fig. 2a and Supplementary Table 2). Most of the dune fields are isolated, and the rest are connected to the Circumpolar Dune Field. Even in the case of S10, we see a dark field directly downslope, exhibiting a similar albedo as the dune field near S9 (Supplementary Fig. 4). We suggest that this dark field is probably a so-called sand patch or protodune that will eventually evolve into dunes[37].

Although all known active scarps are situated above cavi outcrops, the reverse is not always true. In some instances, we see inactive (orange arrow in Supplementary Fig. 1) or no marginal scarps (pink box in Supplementary Fig. 1) above cavi outcrops that are themselves not obviously eroded. Interestingly, there are no dune fields located downslope of these regions either. One interpretation is that erosion of the BU outcrops may be a prerequisite, so that aeolian erosion and the removal of sand from the BU[20,38] occur prior to the onset of ice block falls at the NPLD.

Our results show a strong alignment between the active scarps and the presence of both the cavi outcrops and the interior dune fields, which lends support to the previously proposed hypothesis[8,14,20], that BU erosion is undercutting the NPLD, and the sediments eroded from the cavi unit supply the adjacent dune fields. The conceptual model of the undercutting effect is illustrated in Fig. 4. The NPLD in its history may have been much larger in extent. The quantification of mass wasting activity in this study serves as an indicator of the active BU outcrops that are undergoing erosion. Moreover, this observation also supports the notion that gypsum was released at the active scarps from the BU and possibly from the NPLD, as gypsum has been found within the polar dune fields[6,15,25].

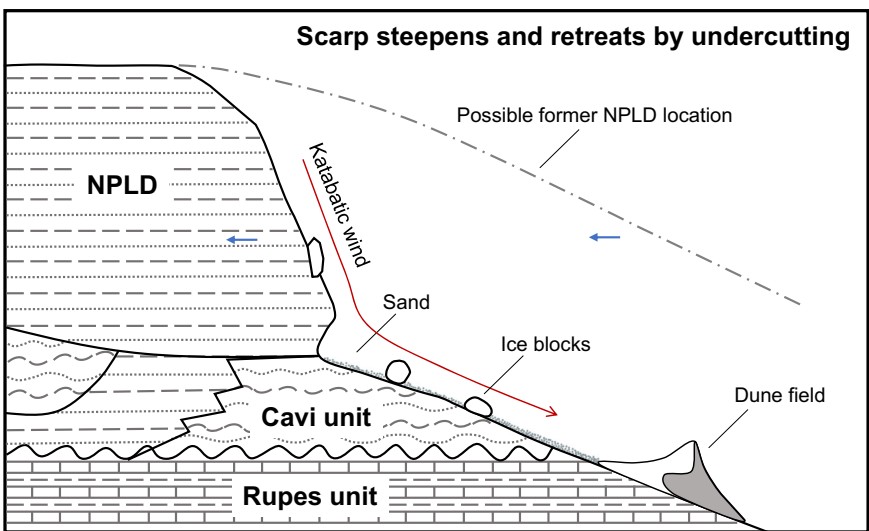

**Fig. 4 | Conceptual model of the undercutting effect.** The schematic diagram illustrates the NPLD scarp base being undermined due to the removal of sand from the BU by katabatic wind, triggering ice block falls and facilitating over-steepened scarp slope over time. Sediments (sand and possibly water ice) are transported to form the dune field nearby. Stratigraphic column is taken from Fig. 3 of the ref. 21.

## Evidence for ongoing asymmetric retreat of the ice cap

Interestingly, there are neither BU outcrops nor isolated interior dune fields in the vicinity of most inactive scarps, like those situated within 300°E–110°E, ~80°N (Supplementary Fig. 1). These scarps may not have been active in the past either because they are not underlain by BU[28]. The BU extent at this longitude range starts at ~85°N, where we found activity between ~330°E and 0°E at Chasma Boreale. Regions with only rupes outcrops (~240°E–300°E) do not seem to trigger activity at the overlying NPLD. At this longitude range, the cavi unit does not reach the latitudes of the marginal scarps and is buried underneath the NPLD further towards the center of the cap[39]. Only two small scarps further south at Abalos Mensa are active, with cavi outcrops and dunes downslope. These areas could have been active also in the past. Overall, active NPLD scarps are found mainly between ~110°E–240°E, ~270°E–300°E and ~330°E–0°E, accounting for only a portion of the north polar ice cap's perimeter. The reason is that their locations are highly correlated with cavi outcrops, which are also scarcely exposed. The only exceptions are three small regions (black arrows in Supplementary Fig. 1) that exhibit cavi outcrops and dune fields while lacking evidence of active scarps. This may be partly due to the absence of observations, but we cannot rule out the possibility that the scarps at these regions were active in the recent past. Overall, the distribution of the active scarps, coupled with the extent of the BU outcrops, indicate a potential ongoing asymmetric retreat of the north polar ice cap, as it is consistent with its current asymmetrical subcircular shape.

Previous studies suggested that the north polar ice cap could have been larger and of more circular shape, probably extending to ~77°N, so that the center of the ice cap was right at the north rotational pole[40,41]. Although our results suggest an ongoing asymmetric retreat of the ice cap, the retreat of the ice cap to where it is today would have proceeded at a much higher pace than our current estimate. The BU outcrops and the interior dune fields at the Olympia Cavi region remain permanently exposed without accumulating ice deposits, distinct from their surrounding plains of Olympia Planum, which are covered by residual ice deposits. These exposed regions are likely the evidence left by the reworking and degradation of the north polar ice cap, indicating that the erosion process might be outpacing accumulation in these regions. Therefore, the mass wasting activity at the marginal scarps probably has been playing a significant role in explaining the current shape and even the long-term evolution of the NPLD.

## Future work

This study shows a systematic way to constantly monitor mass wasting from steep marginal scarps of the north polar ice cap. Meanwhile, it provides supplementary observation basis for the HiRISE camera team. An important task for the future is to use our methods and updated observation data to continuously monitor polar scarps intra-seasonally, inter-seasonally, and annually. This will eventually provide valuable reference data for the upcoming exploration of climate and water resources on Mars.

## Methods
### Datasets

Our study is entirely based on the analysis of HiRISE images from the Mars Reconnaissance Orbiter (MRO). Repeat imaging of the same location with similar illumination conditions facilitates the detection of changes in surface cover and morphology. Since images from the Reduced Data Records (RDRs) do not have the rigorous geometric stability needed for change detection, we instead used images from the Experimental Data Records (EDRs). Each EDR is one of two channels that comprise a charge-coupled device (CCD). The 10 RED CCDs, in total of 20 EDRs, were assembled using the USGS Integrated Software for Imagers and Spectrometers[42,43]. The Digital Terrain Models (DTMs) were produced from stereo pairs using the open-source NASA Ames Stereo Pipeline[44]. We followed the data processing pipeline described previously[16] to produce DTMs and orthorectify the images. Pairs of orthorectified images, used for change detection, are further co-registered for sub-pixel accuracy, following the coarse-to-fine co-registration steps[16]. We use 10 m-scale DTM products for the elevation reference of orthorectification as well as for calculating the slopes of the scarps. The orthorectified images typically have a ground pixel size of 0.25 m/pixel, except for one specific pair of multi-temporal images with 0.5 m/pixel (Supplementary Table 3).

### Ice-fragments detection

Ice-fragments are the sources of ice block falls at the NPLD scarps. The intensity of the ice-fragments in the image is very similar to their surroundings. As a consequence, it is sometimes difficult to identify their entire boundary lines, even by visual inspection. Fortunately, owing to the low-sun conditions in Mars' polar regions, one can usually identify detaching fragments by the change of their cast shadows, which is exploited by artificial intelligence applied in this study. Based on the

above conditions, a deep learning-driven change detection algorithm was designed to detect the detached ice-fragments at steep NPLD scarps by comparing images before and after the detachment[27]. The proposed method used a popular residual convolutional neural network architecture with a customized loss function to target the regions of interest. The method has been proven to be robust and efficient in obtaining reliable results[27]. Visualization examples of the detections by the deep learning method compared to the manual mapping are given in Supplementary Fig. 5. We directly used the trained model[27] to perform change detection. Processing one pair of orthorectified HiRISE images, with a rough size of 9 × 2 km, took around 8 min by using a NVIDIA GeForce GTX 1070 with Max-Q Design. The following 3 factors lead the decision on the selection of change detection pairs, in descending order of priority: 1. the images are well orthorectified; 2. the overlap of the pairs is as large as possible; 3. the time interval of the pairs is as long as possible.

### Ice-fragments' thickness

The detached ice-fragments originate from jagged slab-like fractures, which often open near the surface of the steep scarps, leaving gaps between the fractures and the subsurface. These fractures tend to be sheeting joints rather than being perpendicular to the scarp surface. The conventional ways to determine the thickness of fractures involve deriving height measurements based on their cast shadows or utilizing elevation differences from multi-temporal DTMs. However, these solutions are not feasible because the calculation includes the distance between the opening fracture and the subsurface. To address this challenge, we looked for certain areas where the ice-fragment detached from the scarp and a niche was shown in the 'After' image (Supplementary Fig. 6). The depth of the niche roughly equals to the thickness of the detached ice-fragment.

Our approach is to derive the average thickness of the detached ice-fragments from a collection of niches, similar to the example in Supplementary Fig. 6. Unfortunately, the niches are not always visible in the image, especially those of small ones. We selected visible representative niche examples from 5 scarps, and more specifically 22 different niche samples in various sizes. The shadow, cast by the intact ice around the niche (Supplementary Fig. 6), is used to obtain the depth of the niche (Supplementary Method). The depths, roughly equals to the thicknesses of the ice-fragments, range from 0.7 m to 2.8 m. To assess the accuracy of the depth calculation, in 11 of the 22 niches, we conducted multiple measurements of the same niche on multi-temporal imagery. The standard deviations vary from 0.02 m to 0.15 m, which are small and indicate low uncertainty in depth calculation. The average value (1.5 m) was assigned as the average thickness of the ice-fragments.

Previous studies have indicated that the thickness of ice layers typically ranges from 10 cm to tens of meters[8,45]. It is unreasonable to assume the average thickness of 1.5 m for small-scale detachments. Here we set 2.25 m² as the threshold of the small-scale ice-fragments, which is the area of a square with a side length of 1.5 m. We thus need to additionally calculate an average thickness for such small-scale ice-fragments. We carefully selected 17 examples of small ice-fragments without obvious opening gaps, and approximated their thicknesses based on their cast shadows (Supplementary Method). The thicknesses range from 0.4 m to 1.0 m. Theoretically, the calculated thicknesses are somewhat greater than their true thicknesses because they include the gap between themselves and the subsurface. Nevertheless, due to their small sizes, the discrepancy has minimal impact on the volume calculation. Both the average and the median values are 0.6 m. We thus adopted this value to further calculate the volume of small ice-fragments.

### Ice-fragments' volume

The volume is equal to the area of the ice-fragment multiplied by an estimated average thickness. To derive the actual areas of the ice-fragments we need to consider the slope of the scarp from which they originated. The formula of converting area on the image plane to that on the scarp face is

$$S_{slope} = \frac{S_{plane}}{\cos \theta} \tag{1}$$

where $S_{slope}$ is the area on the slope surface, which is the true ice-fragment's size. $S_{plane}$ is the corresponding projected area on the horizontal plane, which is directly measured from the ortho-rectified HiRISE image. $\theta$ is the slope angle.

In total, we have detected 20,045 ice-fragments from 19 active NPLD scarps. The area of the detached ice-fragments varies from 0.2 m² to 996.1 m². Note that we have excluded the ice-fragments with an area of less than 3 pixels. The log-log plot of size-frequency distribution shows that smaller-sized ice-fragments (area ≤ 2.25 m²) account for the vast majority (Supplementary Fig. 7a). Since we were not able to derive the thickness of each ice-fragment, average values from 39 manually measured samples were used. These samples are representative both of the size-frequency distribution of the detected ice-fragments, as well as of the visual characteristics of them. For example, we followed strict criteria to select small-scale ice-fragments that are not heavily open from the subsurface, and large ice-fragments with their niches' surrounding ice intact instead of being open from the subsurface. This minimized the gaps between the fractures and the subsurface in order to obtain more precise thickness. As described above, large ice-fragments (area > 2.25 m²) are assumed to have an estimated average thickness of 1.5 m, and small-scale ice-fragments (0.2 m² < area ≤ 2.25 m²) have an estimated average thickness of 0.6 m. The graphs in Supplementary Fig. 7b,c show that the thicknesses of the ice-fragments do not strictly increase with area, but are concentrated at ~1.5 m for the large ones, and at ~0.6 m for the small-scale ones. If we would divide the large ice-fragments into two groups: area above and below 25 m² and calculate the average thicknesses of each group according to Supplementary Fig. 7c, the total volume of the detected ice-fragments would increase by approximately 348.8 m³, accounting for only ~0.4% of the total estimated volume. Therefore, the value 1.5 m is reasonable.

Besides, note that the detected area of the ice-fragment is an approximation as the break line of the ice-fragment is not always clear in the image, and even by visual detection we could only obtain its approximate size, let alone by machine detection[27]. These factors lead to an approximate value for the volume of the ice-fragment. The limited image resolution excludes the amount of the ice-fragments with areas smaller than 3 pixels, which in turn could cause an underestimation of the total volume. In addition, if the slab-like ice-fragments get thinner towards the edges rather than having a relatively constant thickness, the volume calculation should be considered a minimum value, as the depth of the niche was measured at its edge. Overall, our obtained volume of the ice-fragments should be considered as a minimum value.

### Erosion rate and retreat rate

The erosion rate is given as

$$R_{erosion} = \frac{V}{L} \tag{2}$$

where $V$ is the volume of the detached ice-fragments, with the unit of m³/martian year. $L$ is the corresponding scarp length. The unit of erosion rate is m³/martian year per scarp meter.

The retreat rate is given as

$$R_{retreat} = \frac{V}{A} \tag{3}$$

where $A$ is the corresponding scarp area. The unit of the retreat rate is m/kyr. Kyr means 1000 Earth years.

## Data availability

The HiRISE data used in this study is publicly available at the Planetary Data System: https://hirise-pds.lpl.arizona.edu/PDS/. The information of all HiRISE images used for detecting ice-fragments is listed in Supplementary Table 3. The detection results of ice-fragments is available at: https://zenodo.org/records/14265216. The generated volume of ice-fragments and erosion rate of each scarp are given in Supplementary Table 2. The data generated in this study are provided in the Source Data file. Source data are provided with this paper.

## Code availability

The deep learning codes used for detecting ice-fragments can be accessed here: https://doi.org/10.5281/zenodo.14265964.

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

## Acknowledgements

We acknowledge the available HiRISE/CTX data at NASA Planetary Data System, the open-source NASA Ames Stereo Pipeline, and the USGS Integrated Software for Imagers and Spectrometers. We would like to thank Dr. Marion Massé (Nantes Université) for sharing the data of marginal scarps and dune fields with us. We would like to thank Dr. Philipp Gläser (Technical University of Berlin) for giving thoughtful reviews of this paper.

## Author contributions

S.S., L.F., and J.O. conceived and designed the experiments. S.S. carried out all the experiments. H.X. helped S.S. in performing the data analysis. S.S. took the lead in writing the paper and all authors contributed to reviewing and editing the paper. E.H. provided critical comments related particularly to geological controls on the erosion processes of the martian north polar region.

## Fundnig

## Competing interests

The authors declare no competing interests.
