## [Peer Review file · Nature Communications]

Mass wasting reveals ongoing asymmetric retreat of the martian north polar ice cap

Corresponding Author: Dr Shu Su

Personal details present in this Peer Review File have been redacted.

Version 0:

Reviewer comments:

Reviewer #1

(Remarks to the Author)

This manuscript provides measurements of mass wasting at steep cliffs around the perimeter of the north polar cap of Mars, and discusses implications for the evolution of the polar cap as a result of the high observed amounts of mass wasting. These cliffs are an extremely interesting set of features, and are one of the most active places in the inner Solar System outside of Earth. They're certainly worthy of study.

The method is innovative and the results and maps here are a useful contribution to the literature. Generally, I think the paper is well written. I do think there need to be improvements made in the "big picture" of the paper's conclusions and implications before publication, in two major ways.

First, the manuscript should be very clear and specific about the advancement made over Su et al. (2023). For example, at line 110, the manuscript states "Our retreat rate surpasses that of Fanara et al by an order of magnitude. The discrepancy has been discussed by Su et al..." Up to that point in the paper, I had been thinking that the greater retreat rate in this manuscript was the main new result here. If the main results here confirm previous results with a new method, that's OK, please just make sure to be clear about that – I find it a little ambiguous in the current text.

Second, I think that the results here should be more quantitatively and specifically linked to the large-scale evolution of the NPLD. Example questions that should be addressed (or discussed) include: Why are the active scarps only along a small portion of the perimeter of the cap? Did the other portions used to be active long ago? How does the volume of material lost compare to published ideas in the literature for the accumulation rate of the cap – is this a substantial contributor to the overall mass balance? If the cap once extended to 77 N (line 173), how long did it take the cap to go from 77 N to its current extent, given the retreat rates calculated in this paper? Is this consistent with the idea that the NPLD is about 5 Myr, or would it require an older age? The authors are well-positioned to comment on these issues with their work. The last sentence of the abstract states "our study provides important constraints to the present-day mass flux and the topographic evolution of the north polar ice cap in general" – I'm asking to be more specific about what those constraints are.

The surface liquid water idea is a big claim and I believe the discussion (around line 132) needs expansion if it is going to be kept, perhaps with calculations showing the plausibility. Would there expected to be any direct observational evidence of liquid water, given how rapid of a process these mass wasting events are? When is the window of time at which the eutectic temperature might be reached? What salts are being invoked here? I'm skeptical that thawing into liquid water is more likely than sublimation.

Can the authors provide a small, local case of the machine learning methods giving consistent results of manual analysis? This would give the reader confidence in the method.

I'd recommend just writing out "Martian year" in figure legends and a few other places to avoid confusion with Megayear.

The manuscript can be a little more specific in saying where most of the active scarps are (line 78), it looks more narrow than 90 E to 270 E – maybe 100 E to 240 E or so?

Reviewer #2

(Remarks to the Author)

This systematic study of scarp retreat rates provides a comprehensive database of present retreat rates of steep marginal scarps of the north polar ice cap. It reinforces the current understanding that the steep scarps retreat by undermining by erosion of the basal unit. The methodology seems appropriate and provides a systematic assessment of retreat rates for actively retreating scarps. A particularly useful result is it demonstrates that glacial flow rates are orders of magnitude lower than some predicted model estimates. The paper is well-written and the conclusions seem justified.

A major caveat is that retreat rate estimates are based on a short time scale (a few martian years) and are only characteristic of the present epoch in the dramatic quasi-periodic climate variation of Mars, which has strong variability over the probable lifetime of the steep polar scarps, particularly at the precession timescale.

A small comment: Please indicate the latitude range represented by Fig. 2a, perhaps by tic marks.

Reviewer #3

(Remarks to the Author)

In this manuscript, Su et al. derive a comprehensive map of mass wasting over the North Polar Layered Deposits (NPLD) on Mars, using an AI-based method to analyze high-resolution images. They find that mass wasting is more active than previously thought, and they estimate scarp retreat rates of up to 3m per kyr due to this process. The results reveal a spatial distribution of scarps with active mass wasting processes that correlates with exposures of the Basal Unit (BU), providing evidence that erosion of the BU undermines the NPLD. The location of the active scarps supports that gypsum was released at the active scarps from the BU, and suggests an ongoing asymmetric retreat of the NPLD and that mass wasting processes plays a role in explaining the current shape of the NPLD.

Overall, the manuscript presents convincing evidence of an active erosion process of the NPLD and provide an interesting explanation to how the current asymmetric shape of the NPLD has formed. These findings are interesting and provide important constraints for understanding the evolution of ice on Mars, which makes the results of wider interest to a broader audience.

The method is developed based on previous work (Su et al. 2020, 2023; Fanara et al. 2020). The main results are the first detection of the active scarps across the NPLD, combined with a new method to estimate the erosion rates. The AI-based method to detect the active scarps are presented in previous work (Su et al. 2023) and is based on pairs of orthorectified images and Digital Terrain Models (DTMs) (Su et al. 2020), and is here for the first time here used for the entire NPLD. This step is robust and well argued, and the detection of active scarps across the NPLD and their spatial patterns are new and interesting results.

The erosion rates are here estimated by using an average thickness of the eroded ice-fragments for large and small (area < 2.25 m²) ice-fragments, respectively, estimated from a random selection of 22 large and 17 small ice-fragments, respectively. The thicknesses of the selected ice-fragments were estimated to be the depth of the eroded niche in the image taken after the mass wasting. The method of analyzing the source region of the ice-fragment to estimate their volume seems very convincing. It seems to be an improvement over the previously used method based on identifying ice blocks under block falls and estimate their total volume, where smaller grains cannot be detected, and the method therefore underestimates the erosion rates.

However, while the method is well described, I have some comments about the estimated erosion rates. My first comment is that it is not clear whether the depth of the niche is actually a good estimate for the thickness of the entire ice-fragment. Is it likely that the thickness is relatively constant for the entire ice-fragment? Or could it be thinning towards the edges, in that case the thickness estimate is a maximum estimate. Please provide some insights to this and discuss the associated uncertainty. My second comment is that it is not completely clear whether the randomly selected ice-fragments are representative. How many ice-fragments were typically detected, and what is the size-distribution? It would be interesting to see some statistical information on the identified mass wasting events to make the estimates better argued, and perhaps help assess whether the erosion estimate is actually a minimum estimate, as mentioned. I don't think this information is included in the supplementary material. I suggest that the discussion of the estimated erosion rates and the uncertainties are elaborated to include these comments.

The discussion of the findings is interesting and well referenced. The discussion of the erosion in relation to estimated flow rates provide some new and relevant input to this long lasting discussion of NPLD flow rates. The manuscript is well written with a clear and logical structure. The figures are clear.

I have a few minor points:

Line 14: reorganize sentence to: "Moreover, ice block activity suggests potential areas...."

Line 16-17: replace "for the possibility" with "whether".

Line 21: insert reference for 95% water ice and 5% dust

Reviewer #4

(Remarks to the Author)

In this paper, authors made a research on Mass wasting and revealed ongoing asymmetric retreat of the martian north polar ice cap by the means of AI and remote sensing data. The principal line of the paper is not clear. The presentation of aim and results in this paper is not clear also. More importantly, the novelty of AI change detection techniques is limited. This

manuscript is not publishable. More detailed comments are listed as follows.

1. The technique values are not high, the proposed change detection AI method is nothing new. The reviewer can not see the reasonable details to support the reliable change detection results.
2. The importance for the research topic is not so clear to see the values.
3. What is the main difference between the studied topic and techniques and existing methods?
4. The important values for this paper are missing. The reviewer can not find it.
5. The methodology and assess authenticity need to be further given.
6. The future works are also blurred. This further reduces the values.

Version 1:

Reviewer comments:

Reviewer #1

(Remarks to the Author)

I find that the manuscript satisfactorily addresses my previous review, especially in making the significance of the work more clear. I thank the authors for their careful revision, and I believe that the manuscript will make a very interesting contribution to the literature. I have two very minor additional suggestions below for the authors to consider, but do not think I need to see the manuscript again before publication.

The last sentence of the abstract is somewhat generic. Can it be made more specific and useful? Perhaps something like "Our study reveals the rates of present-day topographic change of the north polar ice cap, providing a valuable constraint for study of its past evolution."

Line 116: I think I know what is meant here, but the wording confused me because it made it sound like HiRISE resolution is the main issue, when really it is looking at source along the scarps vs fallen ice blocks. I recommend spelling out the method again here, i.e. "... and thus undetectable when searching for fallen material alone."

- Mike Sori [redacted]

Reviewer #3

(Remarks to the Author)

I have now read the revised manuscript of Su et al. I find that my points have been sufficiently addressed and my questions regarding the estimated erosion rates have been improved in the revised manuscript. In particular, the new section on the discussion of erosion rates and the new supplementary figure add quantitative information to assess better the estimated erosion rates.

I also find that the conclusions are well supported in the revised manuscript. As already noted in my first review I found that the results are significant and provide important constraints for understanding the evolution of ice on Mars. I have also checked the responses to the other reviewers and found that the responses carefully address the comments. I have no further comments.

We would like to thank the reviewers for the constructive comments and suggestions, which were really helpful in improving our manuscript. In this document you will find our responses to your questions in blue. Our revisions according to your suggestions are highlighted in the revised manuscript in yellow. We hope that our responses and revisions effectively address your concerns.

REVIEWER COMMENTS

Reviewer #1 (Remarks to the Author):

This manuscript provides measurements of mass wasting at steep cliffs around the perimeter of the north polar cap of Mars, and discusses implications for the evolution of the polar cap as a result of the high observed amounts of mass wasting. These cliffs are an extremely interesting set of features, and are one of the most active places in the inner Solar System outside of Earth. They're certainly worthy of study.

The method is innovative and the results and maps here are a useful contribution to the literature. Generally, I think the paper is well written. I do think there need to be improvements made in the "big picture" of the paper's conclusions and implications before publication, in two major ways.

We appreciate the reviewer's insightful comment and agree that it is necessary to improve the manuscript's conclusions and implications.

First, the manuscript should be very clear and specific about the advancement made over Su et al. (2023). For example, at line 110, the manuscript states "Our retreat rate surpasses that of Fanara et al by an order of magnitude. The discrepancy has been discussed by Su et al..." Up to that point in the paper, I had been thinking that the greater retreat rate in this manuscript was the main new result here. If the main results here confirm previous results with a new method, that's OK, please just make sure to be clear about that – I find it a little ambiguous in the current text.

Thank you for this comment! It helped us clarify what we actually mean here and avoid misunderstandings. The greater retreat rate is a completely new result that we were not able to reach with the method of Su et al. (2023). That method could only detect the casting shadows and not the detached ice-fragments themselves, leading to only a rough estimation of the erosion rate. More specifically, **the estimation from Su et al. (2023) was very similar to that of Fanara et al. (2020)**, which already surprised us, as we expected it to be lower than that of Fanara et al. (2020) since we were not considering the BU activity, but only the NPLD one. This already gave us an indication that it is worthwhile to look at the source of the activity, as it may reveal a volume that cannot be detected in the resulting block falls, i.e., that of the smaller particles. **In this study, with a higher method accuracy than ever, we find a greater retreat rate and indeed by an order of magnitude**, proving our speculation. In order to make our statements clearer, we have changed the corresponding sentences into: "Nevertheless, our retreat rate surpasses that of Fanara et al.⁹ by an order of magnitude. The discrepancy most likely reflects the fact that ice-fragments breaking into ice blocks also produce an additional, substantial quantity of fine and pulverized materials that are not visible at the HiRISE scale, and thus are undetectable by the method used by Fanara et al.⁹." The changes can be found in page 5, lines 114–116 of the revised manuscript. **We also moved the Su et al. (2023) citation to the next sentence where we state the importance of looking at the source region, as this is something we had already stated in that publication.**

Here is the text from the Su et al. (2023) publication we are referring to: *"This corresponds to a minimum erosion rate of 0.2–0.4 m³ per Mars year per meter along the scarp. Fanara et al. (2020) estimated for the same scarp and time interval, but based on the volume of ice block falls on the BU, an erosion rate of ~0.3 m³ per Mars year per meter along the scarp. Our estimated erosion rate takes into account only the NPLD scarp region, whereas the erosion rate calculated by Fanara et al. (2020) represents activity from both geological units, i.e., NPLD and BU, so one would expect the former to be lower. However, our detections include the volume fraction that does not result in block falls, but in rather finer material, not clearly visible at the foot of the scarp and thus not detectable by the method of Fanara et al. (2020) which could lead to larger erosion estimation. This shows the importance of and the need for a more accurate calculation of the source areas' volume, to differentiate between the various volumes and reach a more conclusive result."*

References:

Su, S., Fanara, L., Zhang, X., Hauber, E., Gwinner, K. & Oberst, J. Searching for the sources of ice block falls at the martian north polar scarps. *Icarus* 390, 115321 (2023).

Fanara, L., Gwinner, K., Hauber, E. & Oberst, J. Present-day erosion rate of north polar scarps on Mars due to active mass wasting. *Icarus* 342, 113434 (2020).

Second, I think that the results here should be more quantitatively and specifically linked to the large-scale evolution of the NPLD. Example questions that should be addressed (or discussed) include: Why are the active scarps only along a small portion of the perimeter of the cap? Did the other portions used to be active long ago?

Thank you for your suggestions. The active NPLD scarps are highly correlated with the underlying cavi outcrops (section “NPLD activity is triggered by BU undercutting”), which are exposed at a small portion of the perimeter of the cap. The BU thickness map derived from radar data (Figure a below, Ojha et al., 2019) shows that:

1. the BU is not at all present underneath the NPLD margins between $\sim 60^{\circ}\text{W}$ – 110°E , $\sim 80^{\circ}\text{N}$ in a counterclockwise direction. The BU extent at this longitude range starts at around $\sim 85^{\circ}\text{N}$, where we found activity between $\sim 0^{\circ}$ – 30°W at Chasma Boreale.

[figure redacted]

Figure a, Radar isopach maps showing the thickness of the basal unit (BU). The white outline shows the geographical extent of the north polar ice cap. (Ojha et al., 2019)

2. between $\sim 60^{\circ}\text{W}$ – 120°W the BU is present underneath the NPLD margins, but in the form of rupes unit and not cavi unit. According to a radar data-based study (Figure b and c below, Nerozzi et al., 2022), the cavi unit does not reach those latitudes at this longitude range and is buried underneath the NPLD further towards the center of the cap. Only two small scarps further south at Abalos Mensa are active, with cavi outcrops and dunes downslope (Fig. 2a in the manuscript).

[figure redacted]

Figure b, Topographic map of the BU and its extent is outlined in pink. (Nerozzi et al., 2022)

[figure redacted]

Figure c, (a) Sample of SHARAD profile 600002 and (b) interpretation cartoon showing erosional southward-facing scarps at the top of the rupēs unit and a possible impact crater structure within the cavi unit and the lower section of the NPLD. The location of the profile is shown in Figure b. (Nerozzi et al., 2022)

Therefore, the reason the active scarps (Fig. 2a in the manuscript) are only between $\sim 110^{\circ}\text{E}$ – 240°E (or 110°E – 120°W), $\sim 270^{\circ}\text{E}$ – 300°E (or 60°W – 90°W) and $\sim 330^{\circ}\text{E}$ – 0°E (or 0° – 30°W) is that this is where the cavi unit is exposed underneath the margin of the NPLD. Interestingly, these regions align well with the asymmetrical subcircular shape of the ice cap, showing evidence of ongoing asymmetric retreat (section “Evidence for ongoing asymmetric retreat of the ice cap”).

With respect to the second question on whether other portions used to be active long ago, we are afraid we cannot answer it confidently based on a couple of decades worth of data. However, if we were to hypothesize based on the current observations as stated above, we would argue that only some steep scarps that connect to currently active ones directly above cavi outcrops would have been active in the past. In addition, the

single scarps with both cavi outcrops and dune fields downslope, and at similar latitude with other active ones, may have been active in the past (e.g., the ones pointed by the black arrows in Supplementary Fig. 1).

We have added the corresponding discussion in page 7, lines 171–184 of the revised manuscript:

“Interestingly, there are neither BU outcrops nor isolated interior dune fields in the vicinity of most inactive scarps, like those situated within 300°E–110°E, ~80°N (Supplementary Fig. 1). These scarps may not have been active in the past either because they are not underlain by BU³⁰. The BU extent at this longitude range starts at ~85°N, where we found activity between ~330°E–0°E at Chasma Boreale. Regions with only rupes outcrops (~240°E–300°E) do not seem to trigger activity at the overlying NPLD. At this longitude range, the cavi unit does not reach the latitudes of the marginal scarps and is buried underneath the NPLD further towards the center of the cap²⁸. Only two small scarps further south at Abalos Mensa are active, with cavi outcrops and dunes downslope. These areas could have been active also in the past. Overall, active NPLD scarps are found mainly between ~110°E–240°E, ~270°E–300°E and ~330°E–0°E, accounting for only a portion of the north polar ice cap’s perimeter. The reason is that their locations are highly correlated with cavi outcrops, which are also scarcely exposed. The only exceptions are three small regions (black arrows in Supplementary Fig. 1) that exhibit cavi outcrops and dune fields while lacking evidence of active scarps. This may be partly due to the absence of observations, but we cannot rule out the possibility that the scarps at these regions were active in the recent past. Overall, the distribution of the active scarps, coupled with the extent of the BU outcrops, indicate a potential ongoing asymmetric retreat of the north polar ice cap, as it is consistent with its current asymmetrical subcircular shape.”

References:

Ojha, L., Nerozzi, S. & Lewis, K. Compositional Constraints on the North Polar Cap of Mars from Gravity and Topography. *Geophysical Research Letters* 46, 8671–8679 (2019).

Nerozzi, S., Ortiz, M. R. & Holt, J. W. The north polar basal unit of Mars: An Amazonian record of surface processes and climate events. *Icarus* 373, 114716 (2022).

How does the volume of material lost compare to published ideas in the literature for the accumulation rate of the cap – is this a substantial contributor to the overall mass balance?

Estimations on the accumulation rate of the north polar ice cap are quite different among different modelling methods. For example, Levrard et al. (2007) inferred a positive accumulation rate of 0.9 mm/yr from a global climate model. Later on, Becerra et al. (2017) applied a climate model to match the NPLD stratigraphy and inferred a net accumulation rate of 0.5 mm/yr. Bramson et al. (2019) applied the climate model to match two local polar troughs, thus inferred a net accumulation rate of 0.2 mm/yr, which is a factor of ~4 less than that of Levrard et al. (2007). In addition, Izquierdo et al. (2023) improved the modelling method in Bramson et al. (2019) and got a lower accumulation rate (~0.1 mm/yr) than previously proposed. The above estimated accumulation rates lead to a net mass gain of ~0.2 to 1.8×10^{12} kg/yr. Only considering ice block falls, assuming a density of $1,200 \pm 200$ kg/m³ for the NPLD, the interannual mass wasting of the NPLD is $\sim(9.2 \pm 1.5) \times 10^7$ kg, which cannot compete with the estimated mass gain of the whole NPLD. However, ice ablation is not considered and current modelling methods remain highly uncertain due to lack of constraints. We have added this discussion in page 4, lines 80–86 of the revised manuscript:

“Only considering ice block falls, and assuming a density of $1,200 \pm 200$ kg/m³ for the NPLD³⁰, we estimate the mass wasting rate of the NPLD as $\sim(9.2 \pm 1.5) \times 10^7$ kg/yr, with this number increasing if ice ablation is considered. The present-day accumulation rate of the NPLD has been inferred to vary from ~0.1 to 0.9 mm/yr, with associated net mass gain of ~0.2 to 1.8×10^{12} kg/yr^{31–34}, i.e., greatly exceeding our estimated mass loss. However, estimated accumulation rates based on current data and models remain highly uncertain due to lack of constraints. Nevertheless, our study’s quantification, based on observations of mass loss, could inform the models, thus, its use could extend beyond the results themselves.”

References:

Levrard, B., Forget, F., Montmessin, F. & Laskar, J. Recent formation and evolution of northern Martian polar layered deposits as inferred from a Global Climate Model. *Journal of Geophysical Research: Planets* 112, (2007).

Becerra, P., Sori, M. M. & Byrne, S. Signals of astronomical climate forcing in the exposure topography of

the North Polar Layered Deposits of Mars. *Geophysical Research Letters* 44, 62–70 (2017).

Bramson, A. M., Byrne, S., Bapst, J., Smith, I. B. & McClintock, T. A migration model for the polar spiral troughs of Mars. *Journal of Geophysical Research: Planets* 124, 1020–1043 (2019).

Izquierdo, K. et al. Local ice mass balance rates via Bayesian analysis of Mars polar trough migration. *Journal of Geophysical Research: Planets* 128, 1–18 (2023).

If the cap once extended to 77°N (line 173), how long did it take the cap to go from 77°N to its current extent, given the retreat rates calculated in this paper? Is this consistent with the idea that the NPLD is about 5 Myr, or would it require an older age? The authors are well-positioned to comment on these issues with their work. The last sentence of the abstract states “our study provides important constraints to the present-day mass flux and the topographic evolution of the north polar ice cap in general” – I’m asking to be more specific about what those constraints are.

Indeed, this is a good point, and we agree that it would benefit the manuscript to be more specific. The suggestions will help improving the impact of this manuscript. If we take the retreat rate as a constant value, even using the highest value of ~3 mm/yr, it would have taken about 200 Myr for the ice cap to retreat from 77°N to its current extent. However, as far as we know, constant retreating due to ice block falls may not have happened throughout the history of the north polar ice cap. Moreover, in the past 5 Myr, the orbital parameters of Mars have changed dramatically (Laskar et al., 2002), especially the obliquity, which influences the mass gain or loss of the ice cap. Therefore, retreating from 77°N to the current extent only due to mass wasting may not be a realistic scenario. Although our results suggest an ongoing asymmetric retreat of the ice cap, the rate should have been much higher during the history of the NPLD for it to have retreated to where it is today. The corresponding discussion can be found in page 7, lines 185–188 of the revised manuscript:

“Previous studies suggested that the north polar ice cap could have been larger and of more circular shape, probably extending to ~77°N, so that the center of the ice cap was right at the north rotational pole^{43,44}. Although our results suggest an ongoing asymmetric retreat of the ice cap, the retreat of the ice cap to where it is today would have proceeded at a much higher pace than our current estimate.”

Reference:

Laskar, J., Levrard, B. & Mustard, J. F. Orbital forcing of the martian polar layered deposits. *Nature* 419, 375–377 (2002).

The surface liquid water idea is a big claim and I believe the discussion (around line 132) needs expansion if it is going to be kept, perhaps with calculations showing the plausibility. Would there be expected to be any direct observational evidence of liquid water, given how rapid of a process these mass wasting events are? When is the window of time at which the eutectic temperature might be reached? What salts are being invoked here? I’m skeptical that thawing into liquid water is more likely than sublimation.

This is a fair comment. We agree that the claim of surface liquid water is too bold. The potential for liquid water formation under Mars’ present-day conditions has only been discussed (Fischer et al., 2014; Fischer et al., 2016; Martínez et al., 2017), but not proven. For example, ground temperatures at the Phoenix landing site can be above 200 K from ~4:00 to ~22:00 LMST (Local Mean Solar Time) during northern summer, so that brines could form when getting contact with perchlorate salts that have low eutectic temperatures (Fischer et al., 2016; Martínez et al., 2017). The eutectic temperatures of Mg(ClO₄)₂ and Ca(ClO₄)₂ are ~205 K and ~199 K, respectively (Marion et al., 2010; Martínez et al., 2017). Mg(ClO₄)₂ and Ca(ClO₄)₂ salts have been found at the Phoenix landing site (Hecht et al., 2009). Rich Ca-bearing minerals and gypsum (a hydrated calcium sulfate) have been detected at the north polar dunes fields and exposed Basal Unit areas (Langevin et al., 2005; Massé et al., 2010). However, there is no direct observational evidence of liquid water coming from mass wasting events. We would like to take the reviewer’s suggestion to not keep the discussion of liquid water. Therefore, we deleted the discussion of surface liquid water part in page 5, lines 135–139.

References:

Fischer, E., Martínez, G. M., Elliott, H. M. & Rennó, N. O. Experimental evidence for the formation of

liquid saline water on Mars. *Geophysical Research Letters* 41, 4456–4462 (2014).

Fischer, E., Martínez, G. M. & Rennó, N. O. Formation and persistence of brine on Mars: Experimental simulations throughout the diurnal cycle at the Phoenix landing site. *Astrobiology* 16, 937–948 (2016).

Martínez, G. M., et al. The modern near-surface Martian climate: a review of in-situ meteorological data from Viking to Curiosity. *Space Science Reviews* 212, 295–338 (2017).

Marion, G. M., Catling, D. C., Zahnle, K. J., & Claire, M. W. Modeling aqueous perchlorate chemistries with applications to Mars. *Icarus* 207, 675–685 (2010).

Hecht, M. H., et al. Detection of perchlorate and the soluble chemistry of martian soil at the Phoenix lander site. *Science* 325, 64–67 (2009).

Langevin, Y., Poulet, F., Bibring, J.-P., Bibring, J.-P., Gondet, B. Sulfates in the north polar region of Mars detected by OMEGA/Mars Express. *Science* 307, 1584–1586 (2005).

Massé, M., Bourgeois, O., Le Mouélic, S., Verpoorter, C., Le Deit, L., Bibring, J.-P. Martian polar and circum-polar sulfate-bearing deposits: Sublimation tills derived from the North Polar Cap. *Icarus* 209, 434–451 (2010).

Can the authors provide a small, local case of the machine learning methods giving consistent results of manual analysis? This would give the reader confidence in the method.

Yes, sure. Please check the Supplementary Fig. 5 in the file “Supplementary Information”. We have also added the following sentence in page 8, lines 221–222 of the revised manuscript:

“Visualization examples of the detections by the deep learning method compared to the manual mapping are given in Supplementary Fig. 5.”

I’d recommend just writing out “Martian year” in figure legends and a few other places to avoid confusion with Megayear.

Thank you for your suggestion. We have revised “MY” to “martian year” correspondingly in the manuscript.

The manuscript can be a little more specific in saying where most of the active scarps are (line 78), it looks more narrow than 90 E to 270 E – maybe 100 E to 240 E or so?

We agree. We have revised “90°E to 270°E” to “110°E to 240°E”.

Reviewer #2 (Remarks to the Author):

This systematic study of scarp retreat rates provides a comprehensive database of present retreat rates of steep marginal scarps of the north polar ice cap. It reinforces the current understanding that the steep scarps retreat by undermining by erosion of the basal unit. The methodology seems appropriate and provides a systematic assessment of retreat rates for actively retreating scarps. A particularly useful result is it demonstrates that glacial flow rates are orders of magnitude lower than some predicted model estimates. The paper is well-written and the conclusions seem justified.

We appreciate the reviewer's positive comments.

A major caveat is that retreat rate estimates are based on a short time scale (a few martian years) and are only characteristic of the present epoch in the dramatic quasi-periodic climate variation of Mars, which has strong variability over the probable lifetime of the steep polar scarps, particularly at the precession timescale.

Thank you for your comment. In this study, we provide both the current rates of activity and its distribution. It is only the distribution of the activity that shows a strong link to the past. We agree that we should be very cautious in projecting the current retreat rate to the past, because the orbital parameters of Mars have undergone tremendous changes during the lifetime of the north polar ice cap, resulting in polar insolation that is very different from today (Laskar et al., 2002). There is a hypothesis that the north polar ice cap may have once extended to $\sim 77^\circ\text{N}$ (Fishbaugh and Head, 2000; 2001). Based on this, if we assume our estimate as a constant retreat rate, it would take about 200 Myr for the ice cap to go from 77°N to its current extent, which would be a much longer time than the current estimate of the north polar ice cap's age. Therefore, the retreat rate in the history of the ice cap should have been much higher for it to retreat to today's position. We made this clearer by adding the following sentences in the section "Evidence for ongoing asymmetric retreat of the ice cap", page 7, lines 185–188:

"Previous studies suggested that the north polar ice cap could have been larger and of more circular shape, probably extending to $\sim 77^\circ\text{N}$, so that the center of the ice cap was right at the north rotational pole^{43,44}. Although our results suggest an ongoing asymmetric retreat of the ice cap, the retreat of the ice cap to where it is today would have proceeded at a much higher pace than our current estimate."

Nevertheless, our current retreat rate outcomes can help improve other studies, e.g., facilitating the calibration of Mars climate models by introducing realistic constraints, thus deciphering conditions in the past.

Furthermore, we have added some thoughts on the possible locations of past scarp activity based on a close correlation between the active scarps and the outcrops of the cavi unit. The corresponding discussion can be found in page 7, lines 171–184 of the revised manuscript:

"Interestingly, there are neither BU outcrops nor isolated interior dune fields in the vicinity of most inactive scarps, like those situated within 300°E – 110°E , $\sim 80^\circ\text{N}$ (Supplementary Fig. 1). These scarps may not have been active in the past either because they are not underlain by BU³⁰. The BU extent at this longitude range starts at $\sim 85^\circ\text{N}$, where we found activity between $\sim 330^\circ\text{E}$ – 0°E at Casma Boreale. Regions with only rupes outcrops ($\sim 240^\circ\text{E}$ – 300°E) do not seem to trigger activity at the overlying NPLD. At this longitude range, the cavi unit does not reach the latitudes of the marginal scarps and is buried underneath the NPLD further towards the center of the cap²⁸. Only two small scarps further south at Abalos Mensa are active, with cavi outcrops and dunes downslope. These areas could have been active also in the past. Overall, active NPLD scarps are found mainly between $\sim 110^\circ\text{E}$ – 240°E , $\sim 270^\circ\text{E}$ – 300°E and $\sim 330^\circ\text{E}$ – 0°E , accounting for only a portion of the north polar ice cap's perimeter. The reason is that their locations are highly correlated with cavi outcrops, which are also scarcely exposed. The only exceptions are three small regions (black arrows in Supplementary Fig. 1) that exhibit cavi outcrops and dune fields while lacking evidence of active scarps. This may be partly due to the absence of observations, but we cannot rule out the possibility that the scarps at these regions were active in the recent past. Overall, the distribution of the active scarps, coupled with the extent of the BU outcrops, indicate a potential ongoing asymmetric retreat of the north polar ice cap, as it is consistent with its current asymmetrical subcircular shape."

References:

Laskar, J., Levrard, B. & Mustard, J. F. Orbital forcing of the martian polar layered deposits. *Nature* 419, 375–377 (2002).

Fishbaugh, K. E. & Head, J. W. North polar region of Mars: Topography of circumpolar deposits from Mars Orbiter Laser Altimeter (MOLA) data and evidence for asymmetric retreat of the polar cap. *Journal of Geophysical Research: Planets* 105, 22455–22486 (2000).

Fishbaugh, K. E. & Head, J. W. Comparison of the North and South Polar Caps of Mars: New observations from MOLA data and discussion of some outstanding questions. *Icarus* 154, 145–161 (2001).

A small comment: Please indicate the latitude range represented by Fig. 2a, perhaps by tic marks.

Thanks for your advice. We have used dark gray lines to indicate the latitude ranges and labelled “85°N” and “80°N” at longitude 300°E.

Reviewer #3 (Remarks to the Author):

In this manuscript, Su et al. derive a comprehensive map of mass wasting over the North Polar Layered Deposits (NPLD) on Mars, using an AI-based method to analyze high-resolution images. They find that mass wasting is more active than previously thought, and they estimate scarp retreat rates of up to 3m per kyr due to this process. The results reveal a spatial distribution of scarps with active mass wasting processes that correlates with exposures of the Basal Unit (BU), providing evidence that erosion of the BU undermines the NPLD. The location of the active scarps supports that gypsum was released at the active scarps from the BU, and suggests an ongoing asymmetric retreat of the NPLD and that mass wasting processes plays a role in explaining the current shape of the NPLD.

Overall, the manuscript presents convincing evidence of an active erosion process of the NPLD and provide an interesting explanation to how the current asymmetric shape of the NPLD has formed. These findings are interesting and provide important constraints for understanding the evolution of ice on Mars, which makes the results of wider interest to a broader audience.

Thank you very much for your interest in our work!

The method is developed based on previous work (Su et al. 2020, 2023; Fanara et al. 2020). The main results are the first detection of the active scarps across the NPLD, combined with a new method to estimate the erosion rates. The AI-based method to detect the active scarps are presented in previous work (Su et al. 2023) and is based on pairs of orthorectified images and Digital Terrain Models (DTMs) (Su et al. 2020), and is here for the first time here used for the entire NPLD. This step is robust and well argued, and the detection of active scarps across the NPLD and their spatial patterns are new and interesting results.

The erosion rates are here estimated by using an average thickness of the eroded ice-fragments for large and small (area < 2.25 m²) ice-fragments, respectively, estimated from a random selection of 22 large and 17 small ice-fragments, respectively. The thicknesses of the selected ice-fragments were estimated to be the depth of the eroded niche in the image taken after the mass wasting. The method of analyzing the source region of the ice-fragment to estimate their volume seems very convincing. It seems to be an improvement over the previously used method based on identifying ice blocks under block falls and estimate their total volume, where smaller grains cannot be detected, and the method therefore underestimates the erosion rates.

However, while the method is well described, I have some comments about the estimated erosion rates. My first comment is that it is not clear whether the depth of the niche is actually a good estimate for the thickness of the entire ice-fragment. Is it likely that the thickness is relatively constant for the entire ice-fragment? Or could it be thinning towards the edges, in that case the thickness estimate is a maximum estimate. Please provide some insights to this and discuss the associated uncertainty.

Good question. Thank you for pointing this out. Actually, we measure the depth of the niche at its edge (see the white arrow pointing to the cast shadow in Supplementary Fig. 6), which is also the edge of the detached ice-fragment. If the thickness of the ice-fragment gets thinner towards the edges, the volume estimation should be a minimum value. We have added this discussion in page 9, lines 278–280 of the revised manuscript:

“In addition, if the slab-like ice-fragments get thinner towards the edges rather than having a relatively constant thickness, the volume calculation should be considered a minimum value, as the depth of the niche was measured at its edge.”

My second comment is that it is not completely clear whether the randomly selected ice-fragments are representative. How many ice-fragments were typically detected, and what is the size-distribution? It would be interesting to see some statistical information on the identified mass wasting events to make the estimates better argued, and perhaps help assess whether the erosion estimate is actually a minimum estimate, as mentioned. I don't think this information is included in the supplementary material. I suggest that the discussion of the estimated erosion rates and the uncertainties are elaborated to include these comments.

Thank you for your suggestions. We have detected in total 20,045 ice-fragments from 19 active NPLD scarps across the north polar region. The log-log plot of size-frequency distribution is presented below, which shows that smaller-sized ice-fragments (area ≤ 2.25 m²) account for the vast majority (Figure a). For the thickness

estimation, we have selected 17 small-scale ice-fragments ($0.2 \text{ m}^2 < \text{area} \leq 2.25 \text{ m}^2$) and 22 large ice-fragments ($\text{area} > 2.25 \text{ m}^2$). They are representative of the actual detected ones both in terms of number, as they follow the size-frequency distribution and in terms of visual characteristics, as we have strict selection criteria. More specifically, as we mentioned in the manuscript, the detached ice-fragments originate from fractures, which often open near the surface of the steep scarps, leaving gaps between the fractures and the subsurface. To accurately measure the thickness, we selected samples with minimal openings. We chose the small-scale ice-fragments that are not heavily open from the subsurface, and the large ice-fragments whose eroded niches' surrounding ice is intact and not open from its subsurface.

The thicknesses of the selected 17 small-scale ice-fragments with $0.2 \text{ m}^2 < \text{area} \leq 2.25 \text{ m}^2$ do not strictly increase with their areas, but are concentrated at the average value of $\sim 0.6 \text{ m}$ (Figure b). As we have already discussed in page 8, lines 246–250 of the manuscript, we make sure to pick up the ones without obvious openings, in order to minimize their distance from the subsurface. Unfortunately, we are unable to quantify the extent of the gaps. Setting 0.6 m as the average thickness may lead to an overestimation of the volume, but the discrepancy can be ignored for such small-scale ice-fragments.

Same for the selected 22 large ice-fragments ($\text{area} > 2.25 \text{ m}^2$), we plot their thicknesses versus their areas (Figure c). The graph shows that the ice-fragments with $2.25 \text{ m}^2 < \text{area} \leq 25 \text{ m}^2$ tend to have a thickness of below 1.5 m , with an average thickness of 0.98 m . And, the ice-fragments with an $\text{area} > 25 \text{ m}^2$ tend to have a thickness of above 1.5 m , with an average thickness of 1.89 m . However, please note that the thicknesses of the ice-fragments do not strictly increase with their areas, but are concentrated at the average value of $\sim 1.5 \text{ m}$. The total area of the detected ice-fragments with $2.25 \text{ m}^2 < \text{area} \leq 25 \text{ m}^2$ is $\sim 63008.4 \text{ m}^2$, and the total area of the ice-fragments with an $\text{area} > 25 \text{ m}^2$ is $\sim 84905.6 \text{ m}^2$. To test if the average thickness of 1.5 m is representative, we do the following calculations: for the ice-fragments with $2.25 \text{ m}^2 < \text{area} \leq 25 \text{ m}^2$, the volume will be overestimated by $(1.5 - 0.98) \times 63008.4 \sim 32764.4 \text{ m}^3$; for the ice-fragments with an $\text{area} > 25 \text{ m}^2$, the volume will be underestimated by $(1.89 - 1.5) \times 84905.6 \sim 33113.2 \text{ m}^3$. In total, we underestimate the total volume by $\sim 348.8 \text{ m}^3$, accounting for only 0.4% of the total estimated volume. Therefore, our randomly selected examples are representative for deciding the average thickness of the ice-fragments as 1.5 m .

We would like to take the reviewer's suggestion, and have included the discussion in page 9, lines 258–272 of the revised manuscript:

“In total, we have detected 20,045 ice-fragments from 19 active NPLD scarps. The area of the detached ice-fragments varies from 0.2 m^2 to 996.1 m^2 . Note that we have excluded the ice-fragments with an area of less than 3 pixels. The log-log plot of size-frequency distribution shows that smaller-sized ice-fragments ($\text{area} \leq 2.25 \text{ m}^2$) account for the vast majority (Supplementary Fig. 7a). Since we were not able to derive the thickness of each ice-fragment, average values from 39 manually measured samples were used. These samples are representative both of the size-frequency distribution of the detected ice-fragments, as well as of the visual characteristics of them. For example, we followed strict criteria to select small-scale ice-fragments that are not heavily open from the subsurface, and large ice-fragments with their niches' surrounding ice intact instead of being open from the subsurface. This minimized the gaps between the fractures and the subsurface in order to obtain more precise thicknesses. As described above, large ice-fragments ($\text{area} > 2.25 \text{ m}^2$) are assumed to have an estimated average thickness of 1.5 m , and small-scale ice-fragments ($0.2 \text{ m}^2 < \text{area} \leq 2.25 \text{ m}^2$) have an estimated average thickness of 0.6 m . The graphs in Supplementary Figs. 7b–c show that the thicknesses of the ice-fragments do not strictly increase with area, but are concentrated at $\sim 1.5 \text{ m}$ for the large ones, and at $\sim 0.6 \text{ m}$ for the small-scale ones. If we would divide the large ice-fragments into two groups: area above and below 25 m^2 and calculate the average thicknesses of each group according to Supplementary Fig. 7c, the total volume of the detected ice-fragments would increase by approximately 348.8 m^3 , accounting for only $\sim 0.4\%$ of the total estimated volume. Therefore, the value 1.5 m is reasonable.”

Figure a, The log-log plot of size-frequency distribution of all detected ice-fragments. **b**, Thickness versus area for the selected 17 small-scale ice-fragments ($0.2 \text{ m}^2 < \text{area} \leq 2.25 \text{ m}^2$). **c**, Thickness versus area for the selected 22 large ice-fragments ($\text{area} > 2.25 \text{ m}^2$).

The discussion of the findings is interesting and well referenced. The discussion of the erosion in relation to estimated flow rates provide some new and relevant input to this long lasting discussion of NPLD flow rates. The manuscript is well written with a clear and logical structure. The figures are clear.

Thank you very much for your confirmation!

I have a few minor points:

Line 14: reorganize sentence to: “Moreover, ice block activity suggests potential areas...”

Thanks. Corrections have been done.

Line 16-17: replace “for the possibility” with “whether”.

Thank you for the advice. Here in the abstract, we have removed the phrase: “and raises the question for the possibility of abundant mass wasting to form liquid saline water.” Because the chance of forming liquid saline water from mass wasting remains controversial, and we do not have direct observational evidence yet. Therefore, we have adopted the suggestion of Reviewer #1 to remove the discussion of liquid water.

Line 21: insert reference for 95% water ice and 5% dust

Ok. We have added the following two references:

Malin, M. C. Density of Martian north polar layered deposits: Implications for composition. *Geophysical Research Letters* 13, 444–447 (1986).

Grima, C. et al. North polar deposits of Mars: Extreme purity of the water ice. *Geophysical Research Letters* 36, (2009).

Reviewer #4 (Remarks to the Author):

In this paper, authors made a research on Mass wasting and revealed ongoing asymmetric retreat of the martian north polar ice cap by the means of AI and remote sensing data. The principal line of the paper is not clear. The presentation of aim and results in this paper is not clear also. More importantly, the novelty of AI change detection techniques is limited. This manuscript is not publishable. More detailed comments are listed as follows.

1. The technique values are not high, the proposed change detection AI method is nothing new. The reviewer can not see the reasonable details to support the reliable change detection results.

We are sorry for the reviewer's doubt. However, we would like to point out that this manuscript is not meant to propose a new change detection method. Our AI method was peer-reviewed and published in 2023: *Su, S., Fanara, L., Xiao, H., Hauber, E. & Oberst, J. Detection of detached ice-fragments at martian polar scarps using a convolutional neural network. IEEE Journal of Selected Topics in Applied Earth Observations and Remote Sensing 16, 1728–1739 (2023)*. It has been tested to be robust in detecting mass wasting, and has shown advantages over state-of-the-art deep learning methods. This manuscript is the first large scale application of the already extensively tested and reviewed AI method to detect mass wasting for the entire NPLD.

2. The importance for the research topic is not so clear to see the values.

To address your concern, we would like to clarify the importance of our research topic as follows:

1. Our study presents the first comprehensive database of mass wasting across the entire NPLD on Mars, one of the most active places in the inner solar system outside of Earth.

2. We demonstrate quantitatively that undermining of the NPLD by erosion of the underlying Basal Unit is the main factor triggering the mass wasting processes that are responsible for the asymmetric retreat of the polar cap.

3. Our study results show that the erosion rate is much larger than previously thought and viscous flow rates should be orders of magnitude lower than previously predicted by model estimates.

4. The monitoring and quantification of such amounts of mass loss contributes realistic constraints for understanding the evolution of the north polar ice cap on Mars.

Therefore, we believe that this research topic can bring wider interests to a broader audience. We hope these clarifications can address your concern and strengthen our argument for the importance and values of the research topic.

3. What is the main difference between the studied topic and techniques and existing methods?

The studied topic of mass wasting on Mars' north polar ice cap is a recent finding by the advent of high-resolution images. Existing studies tend to focus on the fallen ice blocks in local scales, while this study is the first one to look for the sources of ice block falls in a regional scale. This is an improvement over existing studies that failed to detect small particles that could not be resolved in the images, thus underestimating the erosion and retreat rates of steep scarps of the north polar ice cap. These statements have been given in our manuscript, in page 5, section "Higher erosion rate than previously thought".

The technique in this study is also different from the existing studies that used a Support Vector Machine (Fanara et al., 2020) or manual mapping (Russell et al., 2014; Herkenhoff et al., 2020). Our proposed deep learning method achieves higher accuracy than Fanara et al., (2020), and is way more efficient than visual detection.

References:

Fanara, L., Gwinner, K., Hauber, E. & Oberst, J. Present-day erosion rate of north polar scarps on Mars due to active mass wasting. *Icarus* 342, 113434 (2020).

Russell, P. S., Feleke, S. & Byrne, S. Landslide erosion rates of north polar layered deposit cliffs and the underlying basal unit. in *Eighth International Conference on Mars* 1373 (2014).

Herkenhoff, K. E., Byrne, S., Dundas, C. M., Baugh, N. F. & Hunter, M. A. HiRISE observations of recent phenomena in the north polar region of Mars. in Seventh Mars Polar Science Conference. (2020).

4. The important values for this paper are missing. The reviewer can not find it.

The important values of this paper are given point-by-point in the Results and Discussion sections, and are summarized in the Abstract. Specifically, this study is the first to provide a comprehensive database of the present-day retreat rates of steep marginal scarps of the north polar ice cap. We show higher values than previously thought. We demonstrate that previously modelled estimates of viscous flow are overestimates. We provide evidence that erosion of the BU undermines the NPLD, thus triggering ice block falls and facilitating over-steepened scarp slope over time. We believe that our research improves the understanding of present-day mass flux dynamics and topographic evolution on Mars.

5. The methodology and assess authenticity need to be further given.

Thank you for your advice. Please find the detailed methodology in the Method section as well as the Supplementary Information file, which explain how we choose and process data, how we perform the AI detection of ice-fragments, how we calculate the thickness and the volume of the ice-fragments, and how we estimate the erosion rate and retreat rate of the scarps. The authenticity of our work is clear if one looks at the extensive literature that we have cited and discussed across the manuscript.

6. The future works are also blurred. This further reduces the values.

We thank the reviewer for the suggestion. We have added the future works at the end of the Discussion section (page 7, lines 194–199 of the revised manuscript:):

“This study shows a systematic way to constantly monitor mass wasting from steep marginal scarps of the north polar ice cap. Meanwhile, it provides supplementary observation basis for the HiRISE camera team. An important task for the future is to use our methods and updated observation data to continuously monitor polar scarps intra-seasonally, inter-seasonally, and annually. This will eventually provide valuable reference data for the upcoming exploration of climate and water resources on Mars.”

We appreciate the positive comments and suggestions from the reviewers. In this document you will find our responses to your questions in blue. Our revisions according to your suggestions are highlighted in the revised manuscript in yellow. We hope that our responses and revisions effectively address your concerns.

REVIEWER COMMENTS

Reviewer #1 (Remarks to the Author):

I find that the manuscript satisfactorily addresses my previous review, especially in making the significance of the work more clear. I thank the authors for their careful revision, and I believe that the manuscript will make a very interesting contribution to the literature. I have two very minor additional suggestions below for the authors to consider, but do not think I need to see the manuscript again before publication.

That's very nice encouragement! We appreciate the reviewer's insightful comments.

The last sentence of the abstract is somewhat generic. Can it be made more specific and useful? Perhaps something like "Our study reveals the rates of present-day topographic change of the north polar ice cap, providing a valuable constraint for study of its past evolution."

Thanks for your suggestion. We have revised it accordingly in the abstract, in page 1, lines 16–17.

Line 116: I think I know what is meant here, but the wording confused me because it made it sound like HiRISE resolution is the main issue, when really it is looking at source along the scarps vs fallen ice blocks. I recommend spelling out the method again here, i.e. "... and thus undetectable when searching for fallen material alone."

- Mike Sori [redacted]

We agree with your concerns. We have revised the sentence according to your suggestions, in page 3, line 98.

Reviewer #3 (Remarks to the Author):

I have now read the revised manuscript of Su et al. I find that my points have been sufficiently addressed and my questions regarding the estimated erosion rates have been improved in the revised manuscript. In particular, the new section on the discussion of erosion rates and the new supplementary figure add quantitative information to assess better the estimated erosion rates. I also find that the conclusions are well supported in the revised manuscript. As already noted in my first review I found that the results are significant and provide important constraints for understanding the evolution of ice on Mars. I have also checked the responses to the other reviewers and found that the responses carefully address the comments. I have no further comments.

Thank you very much for your affirmation! We really appreciate that!